# DUAL GRAPH COMPLEMENTARY NETWORK

## ABSTRACT

As a powerful representation learning method on graph data, graph neural networks (GNNs) have shown great popularity in tackling graph analytic problems. Although many attempts have been made in literatures to find strategies about extracting better embedding of the target nodes, few of them consider this issue from a comprehensive perspective. Most of current GNNs usually employ some single method which can commendably extract a certain kind of feature but some equally important features are often ignored. In this paper, we develop a novel dual graph complementary network (DGCN) to learn representation complementarily. We use two different branches, and inputs of the two branches are the same, which are composed of structure and feature information. At the same time, there is also a complementary relationship between the two branches. Beyond that, our extensive experiments show that DGCN outperforms state-of-the-art methods on five public benchmark datasets.

## 1 INTRODUCTION

Although many attempts have been made in literatures to find a better strategy to learn the target node representation, the feature extraction capabilities of most methods are still far from optimal, especially when only a small amount of data is labeled. However, in fact, compared with the expensive and laborious acquisition of labeled data, unlabeled data is much easier to obtain. Therefore, how to learn more useful representations with limited label information is the key direct of representation learning study. Methods of this issue, commonly referred to as semi-supervised learning, which essentially believe that the similar points have similar outputs. Thus, it can properly utilize the consistency of data to make full use of the rich information of unsupervised data.

In the real world, it is common that we have data with specific topological structures which usually called graph data. The graph structure is usually expressed as the connection between nodes. By aggregating the features of neighborhood and performing appropriate linear transformation, graph neural networks (GNNs) can convert graph data into a low-dimensional, compact, and continuous feature space. Nevertheless, most of them only care about a single aggregation strategy, which is counter intuitive: for example, as far as social networks are concerned, the relationship between people is very complex, while, most of the traditional GNNs only consider the single connection between nodes and ignore other implicit information.

In this paper, our work focuses on learning node representations by GNNs in a semi-supervised way. Despite there are already many graph-based semi-supervised learning methods (Kipf & Welling, 2016; Yang et al., 2016; Khan & Blumenstock, 2019), most of them can only find a single relationship between nodes. As a result, some information in unsupervised data is usually ignored. To overcome this problem, we develop a novel dual graph complementary network (DGCN) to extract information from both feature and topology spaces. An intuition of our method is to learn based on disagreement: network performance is largely related to the quality of the graph, which usually emphasizes the relevance of an attribute of instances. So, since we don't know what attributes are most important, we consider both of them in the model design.

Compared with the traditional GNN-based methods, we perform two different aggregate strategies which emphasize different attributes in each branch, one from the perspective of node feature, and the other from the topological structure. Then, to further utilize implicit information, we employ two networks with different structures to extract embedding from input feature. By doing so, nodes'

information can be propagated in different ways. Then, the supervised loss $\ell_{\mathrm{sup}}$ and diversity constraint $\ell_{\mathrm{div}}$ are used to guide the training.

We use two different branches to extract common information in topology and feature spaces. By utilizing disagreements between the two branches, model can gain information that may be ignored by single branch.

To prove the effectiveness of our method, we conducted experiments on five public benchmark datasets.

The contributions of our work are summarized as follows:

- We propose a novel dual graph complementary network (DGCN) to fuse complementary information, which utilizes different graphs to aggregate nodes that are similar in certain attributes in a complementary way.
- By comparing with algorithms that use non-single graphs, it proves that our complementary architecture can extract richer information
- Through extensive evaluation on multiple datasets, we demonstrate DGCN effectiveness over state-of-the-art baselines.

## 2 RELATED WORK

### 2.1 SEMI-SUPERVISED LEARNING

Semi-supervised learning is usually aimed at the case of insufficient data labels. $X \in \mathbb{R}^{n \times d}$ is the feature of input nodes. $Y = [y_{ij}] \in \mathbb{R}^{n \times k}$ is the label matrix, where $k$ is the class number. $y_{ij}$ means that the $i$-th node belongs to the $j$-th class. Then split data points into labeled and unlabeled points. Accordingly, $x_L$ and $x_U$ express a feature of labeled and unlabeled instance, respectively. Moreover, the ground-truth label of the label nodes is available only.

The main objective of semi-supervised learning is to extract supervised information from labeled dataset whilst adequately utilizing data distribution information contained in $X$. There are four categories of semi-supervised learning algorithms:

1. Self-training semi-supervised learning (Lee, 2013): It utilizes high-confidence pseudo labels to expand label set. Ideally, it can continuously improve network performance, but is usually limited by the quality of pseudo labels.
2. Graph-based semi-supervised learning: It propagates information between instances according to edges in graph. It's an inductive learning method, of which the performance mainly depends on the aggregation algorithm.
3. Low-density separation methods (Joachims, 1999): They assume that the decision hyperplane is consistent with the data distribution, so so it should pass through the sparse region of the data.
4. Pretrain semi-supervised learning: such as autoencoder (Vincent et al., 2008; Rifai et al., 2011), trains the model based on reconstruction error and then fine tune it using labeled data.

However, semi-supervised learning tasks prefer to obtain information related to data distribution rather than all information of samples. In this paper, we mainly focus on the graph-based semi-supervised learning.

### 2.2 GRAPH-BASED SEMI-SUPERVISED LEARNING

In addition to features, graph-based semi-supervised learning methods (Kipf & Welling, 2016) represent the topological edge connection between different instances. For many datasets, graph is given as a feature. If the features of the dataset do not contain the relationships between different samples, a graph can also be constructed by measuring the similarity between the features of the instances (Zhu et al., 2003). Actually, the graph is a measure of whether the instances are closely

connected. Then, according to this graph, information exchange between instances can be carried out, so that the information of unlabeled data can be effectively utilized. Network performance is largely related to the quality of the graph. When the attributes emphasized in the graph do not match the expectations of the task objective, misjudgments are often caused. Usually, it is difficult to finding what really matters. The traditional graph-based semi-supervised learning methods usually uses a single graph for node aggregation, which causes a single attribute to be emphatically considered, but when this attribute does not match the task goal, it will mislead the training instead.

## 3 DGCN ARCHITECTURE

In this section, we will present the overall framework of DGCN, see Fig. 1. The main idea of DGCN is that information exchange under the control of graphs emphasizing different attributes can extract more abundant features. To this end, we use two branches to extract information from two inputs at the same time. The node features of these two inputs are the same, the only difference is the graphs that control the information exchange. In addition, in order to further expand the difference between branches, we use a diversity loss $\ell_{\text{div}}$.

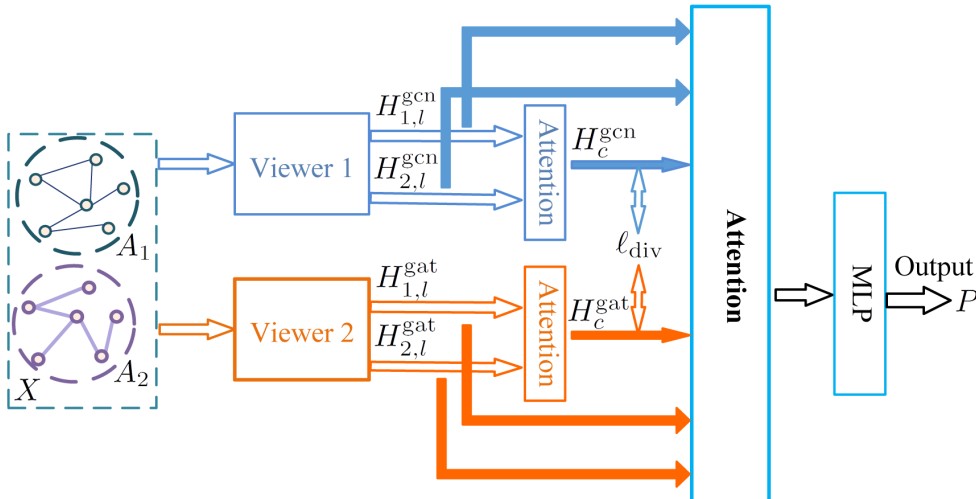

Figure 1: The framework of DGCN network. The original dataset contains the graph and the feature. First, use the node features in the dataset to construct another graph, then use viewer 1 and viewer 2 to observe the two graphs at the same time, and the latent features are $H_{1,l}^{\text{gcn}}$, $H_{2,l}^{\text{gcn}}$, $H_{1,l}^{\text{gat}}$ and $H_{2,l}^{\text{gat}}$ respectively. Then, we fuse GCN view and GAT view respectively to obtain $H_c^{\text{gcn}}$ and $H_c^{\text{gat}}$ respectively through attention operation. The obtained $H_c^{\text{gcn}}$ and $H_c^{\text{gat}}$ are sent to the final attention layer together with the previous $H_{1,l}^{\text{gcn}}$, $H_{2,l}^{\text{gcn}}$, $H_{1,l}^{\text{gat}}$ and $H_{2,l}^{\text{gat}}$.

### 3.1 NOTATION & PROBLEM STATEMENT

Let $\mathcal{G} = (\mathcal{V}, A, \mathcal{X})$ be an undirected graph. $\mathcal{V}$ is the set of nodes on the graph, which is composed of unlabeled ($\mathcal{V}_u$) and labeled ($\mathcal{V}_l$) nodes with the number of nodes is $n_u$ and $n_l$ respectively. $n = n_l + n_u$ is the number of nodes. $A = [a_{ij}] \in \mathbb{R}^{n \times n}$ is the adjacency matrix. $a_{ij} = 1$ represents that node $i$ and node $j$ are closely related in an attribute, otherwise, $a_{ij} = 0$.

### 3.2 BRANCHES

In order to capture different characteristics by the two branches (also called viewer), we use different network structures for each branch: GCN (Kipf & Welling, 2016) and GAT (Veličković et al., 2017). Given a graph $\mathcal{G} = (\mathcal{V}, A, \mathcal{X})$, both GCN and GAT intend to extract richer features at a vertex by aggregating features of vertices from its neighborhood (Li et al., 2019). So the node representation

of the $l$-th layer $H_l$ can be defined by:

$$H_l = \text{Update}(\text{Aggregate}(H_{l-1}, \Theta_l^{\text{agg}}), \Theta_l^{\text{update}}). \tag{1}$$

where $\Theta_l^{\text{agg}}$ and $\Theta_l^{\text{update}}$ are the learnable weights of aggregation and update functions of the $l$-th layer respectively and the initial $H_0 = \mathcal{X}$. The aggregation and update functions are the essential components of GNNs, and obviously the features extracted by different aggregation functions will have certain differences. Thus, we take advantage of two different networks, GCN and GAT, to obtain node representation.

The node features output by the $l$-th GCN layer can be expressed as:

$$H_l = \sigma\left(\left(\tilde{D}^{-\frac{1}{2}}(A + I)\tilde{D}^{-\frac{1}{2}}\right)H_{l-1}W_l\right). \tag{2}$$

where $I \in \mathbb{R}^{n \times n}$ indicates the identity matrix, $A + I$ means adding self-loop in the graph, $\tilde{D}$ is the diagonal degree matrix of $A + I$, and $\sigma(\cdot)$ is the activation function.

It can be seen from equation 2 that GCN aggregates neighbor features by weighting the value of symmetric normalized laplacian.

Next, we introduce the algorithm GAT that uses the attention mechanism to calculate the neighbor weight. Through a learnable coefficient $\boldsymbol{a}$, GAT can assign learnable weights to each neighbor of the node. For node $i$, the weight $\alpha_{ij}$ between it and its neighbor node $j$ can be expressed as:

$$\alpha_{ij} = \frac{\exp\left(\text{LeakyReLU}\left(\boldsymbol{a}^\top[W\boldsymbol{h}_i \| W\boldsymbol{h}_j]\right)\right)}{\sum_{k \in \mathcal{N}_i} \exp\left(\text{LeakyReLU}\left(\boldsymbol{a}^\top[W\boldsymbol{h}_i \| W\boldsymbol{h}_k]\right)\right)}. \tag{3}$$

where $\cdot^\top$ is the transposition operation and $\|$ represents concatenation. Then the forward propagation process of node $v$ in $l$-th layer can be represented as:

$$\boldsymbol{h}_{l,i} = \|_{m=1}^M \sigma\left(\sum_{j \in \mathcal{N}_i} \alpha_{l,ij}^m W_l^m \boldsymbol{h}_{l-1,j}\right). \tag{4}$$

where, $\boldsymbol{h}_{l,i}$ is the embeding of node i in the $l$-th layer. $M$ is the number of independent attention mechanisms. $\sigma$ is activation function of GAT. $\alpha_{ij}^m$ is the normalized attention coefficients computed by the $m$-th attention mechanism , see equation 3. As can be seen from equation 4, the weights GAT assigns to a node's neighbors are learnable. Thus we can assign adaptive weights to different neighbors.

Although these two methods are based on the existence of connection between points as the premise of aggregation. Both the GCN and GAT models we use have their own advantages and disadvantages. The former considers the relationship between nodes (probability conduction matrix), but can't learn neighbor weights dynamically. Although the latter can assign dynamic weights to neighbors, it ignores the influence of degree attribute of node on aggregation. Therefore, using these two branches, we can extract more complementary features from the input.

### 3.3 FORWARD PROPAGATION

In this subsection, we introduce the input used by the network and the specific forward propagation strategy. In order to consider different attributes when aggregating, we use different graphs for training, but adopt the same features. In this paper, the datasets used in our experiment are graph-structured which have two characteristics, one is the feature of the instance itself, which is not affected by other instances, and the other reflects the relationship with other instances.

For example, dataset ACM (Wang et al., 2019) which extracted from ACM dataset contains 3025 papers. It has two properties: one is a bag-of-words that indicates whether the keyword exists, and the other indicates which papers are written by the same author.Obviously, if we only base whether the paper is written by the same author as the basis for aggregation, we will inevitably

ignore the situation where the same author has written different types of papers and the same type of papers belong to different authors, thus mistakenly aggregate articles of different categories together. Therefore, we also construct a graph based on another attribute of the dataset: bag-of-words, so that information can be transferred between instances with similar keywords. In order to measure the similarity of instances' features, we find the cosine similarity between the features of all instances:

$$s_{ij} = \frac{\boldsymbol{x}_i \cdot \boldsymbol{x}_j}{|\boldsymbol{x}_i| \, |\boldsymbol{x}_j|}. \tag{5}$$

where $s_{ij}$ denotes the cosine similarity between the feature $\boldsymbol{x}_i$ of node $i$ and the feature $\boldsymbol{x}_j$ of node $j \in \mathcal{V}$.

Notice that $j \neq i$. For node $i$ , we choose the $t$ largest $s_{ij}$ and let the corresponding $j$ as the neighbors of $i$. Then if $j$ is the neighbor of node $i$, obviously $i$ is the neighbor of node $j$ too. As above-mentioned, we can get a new graph constructed from features. We use $A_1$ and $A_2$ to represent the inherent graph structure of the data and the graph constructed according to the feature, respectively.

Therefor, by inputting $A_1$ and $A_2$ for each branch, we can get four different outputs. according to equation 2 and equation 4 the forward propagation of DGCN can be represented as:

$$H_{v,l}^{\mathrm{gcn}} = \sigma\Big( \big( \tilde{D}_n^{-\frac{1}{2}} (A_v + I) \tilde{D}_n^{-\frac{1}{2}} \big) H_{v,l-1}^{\mathrm{gcn}} \Theta_{v,l} \Big). \tag{6}$$

$$\boldsymbol{h}_{v,l,i}^{\mathrm{gat}} = \|_{m=1}^M \sigma \left( \sum_{j \in \mathcal{N}_i} \alpha_{l,i,j}^m W_{v,l}^m \boldsymbol{h}_{v,l-1,j}^{\mathrm{gat}} \right). \tag{7}$$

where, $v = 1$ represents that the graph is $A_1$, while $v = 2$ corresponding to $A_2$. $\sigma$ and $\sigma$ are the activation function. $\alpha_{v,i,j}^k$ is the normalized attention coefficients. $\Theta_{v,l}$ and $W_{v,l}^k$ are the weights of linear transformations. For the GAT branch, $\boldsymbol{h}_{n,l,i}^{\mathrm{gat}}$ means the representation of node $i$ in the $l$-th layer with the input graph is $A_v$. Similarly, $H_{v,l}^{\mathrm{gcn}}$ corresponds to the $l$-layer embedding matrix of the GCN branch when the input graph is $A_v$.

For these four embeddings, we first use the attention mechanism to combine the embeddings generated by different graphs of the same branch:

$$H_c^{\mathrm{gcn}} = \mathrm{att}(H_{1,l}^{\mathrm{gcn}} \| H_{2,l}^{\mathrm{gcn}}). \tag{8}$$

$$H_c^{\mathrm{gat}} = \mathrm{att}(H_{1,l}^{\mathrm{gat}} \| H_{2,l}^{\mathrm{gat}}). \tag{9}$$

Then, we apply the attention mechanism again to combine $H_{1,l}^{\mathrm{gcn}}$, $H_{2,l}^{\mathrm{gcn}}$, $H_{1,l}^{\mathrm{gat}}$, $H_{2,l}^{\mathrm{gat}}$, $H_c^{\mathrm{gcn}}$ and $H_c^{\mathrm{gat}}$. Through these two attention operations, we can dynamically assign weights to different embedding to find attributes that better match the task goal.

### 3.4 Loss Functions of DGCN

The objective function of DGCN consists of two parts: the supervised loss $\ell_{\mathrm{sup}}$ and the diversity loss $\ell_{\mathrm{div}}$.

#### 3.4.1 Supervised Loss

Given a graph $\mathcal{G} = (\mathcal{V}, A, \mathcal{X})$, as $\mathcal{V} = \mathcal{V}_l \cup \mathcal{V}_u$, the corresponding label of $\mathcal{V}_l$ is $Y_l$. In order to utilize the supervision information, we use the cross-entropy loss function to guide the training:

$$\ell_{\mathrm{sup}} = -\sum_{i \in \mathcal{V}_l} \sum_{j=1}^k y_{ij} \ln p_{ij}. \tag{10}$$

where $y_{ij}$ is the ground-truth label of node $i$ and $p_{ij}$ is the model predicted label. $k$ is the number of classes.

### 3.4.2 DIVERSITY LOSS

In order to further expand the differences between branches and capture richer node features, we use $\mathcal{L}_{div}$ to add a diversity constraint on $H_c^{\text{gcn}}$ and $H_c^{\text{gat}}$. First, we use $L_2$-normalization to normalize $H_c^{gcn}$ and $H_c^{gat}$ output by the attention module. The normalized results are $\hat{H}^{\text{gcn}}$ and $\hat{H}^{\text{gat}}$ respectively. Using the above results, we can capture the similarity of node embedding:

$$\hat{S} = \hat{H}_c^{\text{gcn}}(\hat{H}_c^{\text{gat}})^\top \tag{11}$$

Then, the diversity loss can be defined by:

$$\ell_{\text{div}} = \frac{\sum_{i=1}^n \sum_{j=1}^n \hat{s}_{i,j}}{n^2} \tag{12}$$

where $n$ is the number of nodes. Through this diversity constraint, we can expand the difference between the branches to learn complementary features. Therefore, our final optimization object can be expressed as:

$$\ell_{\text{total}} = (1 - \gamma)\ell_{\text{sup}} + \gamma\ell_{\text{div}} \tag{13}$$

where $\gamma$ is parameter of the disparity constraint terms. Using this objective function, we can optimize the proposed model through back propagation and learn the node embedding for classification.

Table 1: Statistics of the datasets. Refer Section 4.1 for more details.

| Datasets | Nodes | Edges | Classes | Features | Training | $\frac{|\mathcal{V}_l|}{|\mathcal{V}|}$ | Test |
|---|---|---|---|---|---|---|---|
| ACM | 3025 | 13128 | 3 | 1870 | 60/120/180 | 0.020/0.040/0.060 | 1000 |
| UAI2010 | 3067 | 28311 | 19 | 4973 | 380/760/1140 | 0.124/0.248/0.372 | 1000 |
| Citeseer | 3327 | 4732 | 6 | 3703 | 120/240/360 | 0.036/0.072/0.108 | 1000 |
| BlogCatalog | 5196 | 171743 | 6 | 8189 | 120/240/360 | 0.023/0.046/0.069 | 1000 |
| Flickr | 7575 | 239738 | 9 | 12047 | 180/360/540 | 0.024/0.048/0.071 | 1000 |

## 4 EXPERIMENT

### 4.1 DATASETS

For evaluating the effectiveness of DGCN, we evaluate on several semi-supervised classification benchmarks. Following the experimental setup of Wang et al. (2020), we evaluate on five datasets.

- ACM (Wang et al., 2019): This dataset is extracted from the ACM dataset, where the nodes represent the papers, the edges represent that the connected two papers belong to the same author, and the feature is the word bag representation of paper's keywords.
- UAI2010 (Wang et al., 2018): This dataset has 3067 nodes and 19 classes.
- Citeseer (Kipf & Welling, 2016): In the CiteSeer dataset, papers are divided into six categories, containing a total of 3312 papers, which record citation information between papers. And the feature is the word bag representation of the paper.
- BlogCatalog (Meng et al., 2019): This is a network of social relationships from the Blog-Catalog websitewhere the nodes are bloggers and edges are their social relationships. Node attributes are the short descriptions of users' blogs provided by users. The labels represent the topic categories provided by the authors which can be divided into 6 classes.
- Flickr (Meng et al., 2019): This network is built from profile and relation data of users in Flickr. We treat each user as a node, relationships between two user as an edge.The labels represent the interest groups of the users.

The detailed descriptions of the datasets used here are shown in Table 1.

Table 2: Experiments results (%) on the node classification task. L/C means the number of labeled nodes per class

| Datasets | L/C | 20 | | 40 | | 60 | |
|---|---|---|---|---|---|---|---|
| | Metrics | ACC | F1 | ACC | F1 | ACC | F1 |
| ACM | DeepWalk | 62.69 | 62.11 | 63.00 | 61.88 | 67.03 | 66.99 |
| | LINE | 41.28 | 40.12 | 45.83 | 45.79 | 50.41 | 49.92 |
| | GCN | 87.80 | 87.82 | 89.06 | 89.00 | 90.54 | 90.49 |
| | kNN-GCN | 78.52 | 78.14 | 81.66 | 81.53 | 82.00 | 81.95 |
| | GAT | 87.36 | 87.44 | 88.60 | 88.55 | 90.40 | 90.39 |
| | DEMO-Net | 84.48 | 84.16 | 85.70 | 84.83 | 86.55 | 84.05 |
| | MixHop | 81.08 | 81.40 | 82.34 | 81.13 | 83.09 | 82.24 |
| | AM-GCN | 90.40 | 90.43 | 90.76 | 90.66 | 91.42 | 91.36 |
| | **DGCN** | **91.10** | **91.07** | **91.40** | **91.35** | **91.90** | **91.90** |
| UAI2010 | DeepWalk | 42.02 | 32.93 | 51.26 | 46.01 | 54.37 | 44.43 |
| | LINE | 43.47 | 37.01 | 45.37 | 39.62 | 51.05 | 43.76 |
| | GCN | 49.88 | 32.86 | 51.80 | 33.80 | 54.40 | 34.12 |
| | kNN-GCN | 66.06 | 52.43 | 68.74 | 54.45 | 71.64 | 54.78 |
| | GAT | 56.92 | 39.61 | 63.74 | 45.08 | 68.44 | 48.97 |
| | DEMO-Net | 23.45 | 16.82 | 30.29 | 26.36 | 34.11 | 29.05 |
| | MixHop | 61.56 | 49.19 | 65.05 | 53.86 | 67.66 | 56.31 |
| | AM-GCN | 70.10 | 55.61 | 73.14 | 64.88 | 74.40 | 65.99 |
| | **DGCN** | **72.50** | **58.57** | **75.80** | **65.89** | **78.00** | **70.19** |
| Citeseer | DeepWalk | 43.47 | 38.09 | 45.15 | 43.18 | 48.86 | 48.01 |
| | LINE | 32.71 | 31.75 | 33.32 | 32.42 | 35.39 | 34.37 |
| | GCN | 70.30 | 67.50 | 73.10 | 69.70 | 74.48 | 71.24 |
| | kNN-GCN | 61.35 | 58.86 | 61.54 | 59.33 | 62.38 | 60.07 |
| | GAT | 72.50 | 68.14 | 73.04 | 69.58 | 74.76 | 71.60 |
| | DEMO-Net | 69.50 | 67.84 | 70.44 | 66.97 | 71.86 | 68.22 |
| | MixHop | 71.40 | 66.96 | 71.48 | 67.40 | 72.16 | 69.31 |
| | AM-GCN | 73.10 | 68.42 | 74.70 | 69.81 | 75.56 | 70.92 |
| | **DGCN** | **74.60** | **69.46** | **75.30** | **71.14** | **76.90** | **72.97** |
| BlogCatalog | DeepWalk | 38.67 | 34.96 | 50.80 | 48.61 | 55.02 | 53.56 |
| | LINE | 58.75 | 57.75 | 61.12 | 60.72 | 64.53 | 63.81 |
| | GCN | 69.84 | 68.73 | 71.28 | 70.71 | 72.66 | 71.80 |
| | kNN-GCN | 75.49 | 72.53 | 80.84 | 80.16 | 82.46 | 81.90 |
| | GAT | 64.08 | 63.38 | 67.40 | 66.39 | 69.95 | 69.08 |
| | DEMO-Net | 54.19 | 52.79 | 63.47 | 63.09 | 76.81 | 76.73 |
| | MixHop | 65.46 | 64.89 | 71.66 | 70.84 | 77.44 | 76.38 |
| | AM-GCN | 81.98 | 81.36 | 84.94 | 84.32 | 87.30 | 86.94 |
| | **DGCN** | **88.70** | **88.31** | **90.30** | **90.02** | **92.00** | **91.69** |
| Flickr | DeepWalk | 24.33 | 21.33 | 28.79 | 26.90 | 30.10 | 27.28 |
| | LINE | 33.25 | 31.19 | 37.67 | 37.12 | 38.54 | 37.77 |
| | GCN | 41.42 | 39.95 | 45.48 | 43.27 | 47.96 | 46.58 |
| | kNN-GCN | 69.28 | 70.33 | 75.08 | 75.40 | 77.94 | 77.97 |
| | GAT | 38.52 | 37.00 | 38.44 | 36.94 | 38.96 | 37.35 |
| | DEMO-Net | 34.89 | 33.53 | 46.57 | 45.23 | 57.30 | 56.49 |
| | MixHop | 39.56 | 40.13 | 55.19 | 56.25 | 64.96 | 65.73 |
| | AM-GCN | **75.26** | **74.63** | 80.06 | 79.36 | 82.10 | 81.81 |
| | **DGCN** | 74.6 | 72.47 | **81.1** | **83.4** | 81.06 | 83.18 |

## 4.2 BASELINES

We compare with some state-of-art baselines to verfify the effectiveness of the proposed DGCN.

- DeepWalk (Perozzi et al., 2014) is a random walk based network embedding method, learning feature by treating truncated random walks in a graph as the equivalent of sentences.
- LINE (Tang et al., 2015) is a large-scale embedding method that retains both the local network structure and the global network structure.
- GCN (Kipf & Welling, 2016) is a variant of convolutional neural networks which aggregates information of nodes to get node characteristics.
- kNN-GCN. The network structure of kNN-GCN is the same as that of GCN. But the graph we use here is the aforementioned $A_2$, see Section 3.3.
- GAT (Veličković et al., 2017) is a graph attention based method which can assign different weights to nodes during aggregation.
- DEMO-Net (Wu et al., 2019) assumes that nodes with the same degree value will share the same graph convolution, and the feature aggregation is expressed as a multi-task learning problem according to the degree value of the node.
- MixHop (Abu-El-Haija et al., 2019) can learn the neighbor mixture relationship by repeatedly mixing the feature representations of neighbors at various distances.
- AM-GCN (Wang et al., 2020) extracts embeddings from node features, topological structures and their combinations, and uses the attention mechanism to learn the adaptive importance weights of embeddings

## 4.3 RESULTS

We train the DGCN network described in Section 3 on five public datasets and evaluate the prediction accuracy on a test set of 1,000 labeled examples, and experiments on all datasets are optimized using Adam optimizer. The model adopts GCN and GAT branches with a layer number of 2, and the quantitative analysis results can be seen in Table 2.

- It can be seen from the Table 2 that DGCN can exceed the baseline in most of the accuracy rates of all datasets, which proves the effectiveness of our method. On most datasets, the performance of DGCN is better than AM-GCN using two graphs and GCN, kNN-GCN, GAT using one graph, which fully proves that DGCN can capture more information that meets the task objectives.In addition, by comparing with AM-GCN, which also uses different graphs for learning, our DGCN can learn better node embeddings through its complementary learning mechanism.
- The main difference between GCN and KNN-GCN is that the structure graph and cosine graph are used respectively. For a dataset, a graph that is more relevant to its classification goal(Nodes connected by edges are more likely to belong to the same class) will perform better. As we can see, on UAI2010, BlogCatelog and Flickr kNN-GCN seem to be significantly better than GCN, but the other two datasets are opposite. This means that on the UAI2010, BlogCatelog and Flickr cosine graph is closer to the classification target than structure graph.

## 5 CONCLUSION

In this paper, aiming at the problem of semi-supervised graph node classification, we proposes a novel dual graph complementary network (DGCN), which can utilize graphs that emphasize the different attributes of the input to guide the aggregation process. In addition, in order to further capture richer information, we use two different branches to perform feature learning separately. At the same time, the disparity constraint is used between the two branches to further expand the difference. However, just using the diversity loss may retain too much unnecessary redundant information, which will interfere with the really important information. Therefore, our future work will try to emphasize the common attributes in the embedding while expanding the differences between

branches. The extensive experiments on several datasets further demonstrate the effectiveness of our DGCN algorithm.

In the future, we will further study the correlation measurement of graphs and training objectives and further enrich our model with them.

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
