# OpenReview forum: "Dual Graph Complementary Network"
_ICLR.cc/2021/Conference — Reject_

### Official Review · AnonReviewer2 · 2020-10-25
**Heuristic idea without much insight, unfair evaluation**

**Rating:** 3
**Confidence:** 5

**Review:**

This work proposes a new method by combining GCN and GAT to perform node semi-supervised classification task. The new model uses node features to build another graph, uses GCN and GAT on the original graph and the new graph, and also adds a loss term to reduce the similarity of the final node representations.

1. The idea is very heuristic without much insight. The paper keeps arguing traditional GNNs only use one-side information, but they are actually leverage both node features and graph structures by propagating nodes features over the graph structure. So the statement is not correct. The paper claims that different node attributes contribute in different ways that should be sufficiently leveraged, but this is a very confusing argument if not paired with empirical justification. In the proposed model, it seems the authors try to resolve the above confusing issue via using another graph structure built only based on only node attributes. There is very unclear connection showing why this method resolves the problem they proposed.

2. The experiment parts are not in a fair comparison too, as the paper does not use the standard way to perform dataset splitting. For this new splitting way, no hyperparameters are report for both the model here and previous models. Some benchmark datasets for semi-supervise learning are also not used, e.g., cora, pubmed,...

3. There are quite a few errors in grammar. I suggest authors to perform some grammar checking.

---

> ### Author Response · Authors · 2020-11-16
> **To Reviewer2**
>
> Thank you for taking the time to review our paper and we appreciate the positive and constructive feedback. In response to some of your concerns, we have made the following responses：
>
> Quote 1: “The paper keeps arguing traditional GNNs only use one-side information, but they are actually leverage both node features and graph structures by propagating nodes features over the graph structure. ”
>
> We have always admitted that traditional GNNs can leverage both node features and graph structures. Actually, the main idea in our paper is that traditional GNNs only uses one graph to
> propagate nodes features. And We believe that graphs also have attributes. Take the citation network as an example. If nodes refer to papers, graph 1 thinks that nodes belonging to the same author should be connected, while graph 2 thinks that nodes with citation relationships should be connected. Obviously, if GNNs use only one graph, some implicit information may be ignored.
>
> Quote 2: “The paper claims that different node attributes contribute in different ways that should be sufficiently leveraged .....”
>
> In fact, as mentioned above, we believe that graph actually reflects whether there is a certain relationship between different nodes (such as citation, same author, etc.). The reason for the success of GNNs lies in their ability to propagating nodes features over the graph structure. And, considering different graphs at the same time may be helpful for us to obtain more sufficient information. Taking the citation network as an example, we believe that if a node can obtain the information of the node belonging to the same author and the information of the node cited by it at the same time, it will get more information.
> But we still need to admit that the graphs we currently use only include the original graph in the dataset and the graph constructed by features. How to get more different graphs is still a problem.
>
>
> Quote 3:“For this new splitting way, no hyperparameters are report for both the model here and previous models. ”
>
> We will submit model parameters in supplementary materials later.
>
> Quote 4:“There are quite a few errors in grammar.”
>
> Thanks for your suggestion, we will check the grammar of the paper.

---

### Official Review · AnonReviewer1 · 2020-10-26
**The work is incremental and straightforward**

**Rating:** 4
**Confidence:** 5

**Review:**

The paper presents a GNN model to jointly encode both topology and feature graphs to enhance node representations' quality. In particular, the model DGCN uses two GCNs to learn and propagate two different types of node representations on the topology graph, respectively. The model also utilizes two GATs to learn and propagate two different types of node representations on the feature graph, respectively. Finally, the model leverages attention mechanisms on these four types of node representations to produce the final node embeddings.

Pros:
+ The model obtains promising results.

Cons:

+ The motivation in the second paragraph of the introduction makes confusion. The quote sentence - "Most of the traditional GNNs only consider the single connection between nodes and ignore other implicit information" - leads to why not considering multiple connections between nodes as there are GNN works on hyper-graphs such as [2].
+ The intuition in the third paragraph of the introduction is not clear as the paper does not have any ablation study for this intuition.
"Network performance is largely related to the quality of the graph, which usually emphasizes the relevance of an attribute of instances", so which references for this intuition? What are the attributes of instances?
+ Using v=1 for A_1 to denote the topology graph and v = 2 for A_2 to denote the feature graph makes the paper harder to read.
+ The paper is not well written as it does not include any descriptions about model parameters in both the paper and the supplementary material. So it's hard to understand how to train DGCN and the baselines and how to analyze the model and ablation studies.
+ The most important one is that, regarding the model architecture, DGCN is precisely similar to AM-GCN. In particular, DGCN changes from using two GCNs for the feature graph in AM-GCN [1] to using two GATs. Therefore, DGCN is straightforward and incremental (i.e., lacking novelty).

[1] AM-GCN: Adaptive Multi-channel Graph Convolutional Networks. KDD 2020.

[2] Hyper-SAGNN: a self-attention based graph neural network for hypergraphs. ICLR 2020.

---

> ### Author Response · Authors · 2020-11-16
> **To Reviewer1**
>
> Thank you for taking the time to review our paper and we appreciate the positive and constructive feedback. In response to some of your concerns, we have made the following responses：
>
> 1、We want to point out that our DGCN uses two different network branches to extract features (GCN and GAT are used in the paper), and the input of these two branches is exactly the same (Extract feature information and structural information at the same time in a forward pass) . Therefore, there is no situation where two GCNs are used to extract structural information and characteristic information and then two GATs are used to extract structural information and characteristic information.
>
> 2、About motivation, the original meaning of "Most of the traditional GNNs only consider the single connection between nodes and ignore other implicit information" is as follows:
> GNNs use graph as the basis of aggregation (that is, if and only if two nodes are connected, they can be aggregated to update representation.). Graph has attributes. Take the citation network as an example. If nodes refer to papers, Graph 1 thinks that nodes belonging to the same author should be connected, while Graph 2 thinks that nodes with citation relationships should be connected. Obviously, if GNNs use only one graph, some implicit information may be ignored. In general, "the single connection" refers to a single graph, and our paper does not use the knowledge of hyper-graphs.
>
> 3、We will submit model parameters in supplementary materials later.

---

### Official Review · AnonReviewer4 · 2020-10-26

**Rating:** 2
**Confidence:** 4

**Review:**

This paper presents a dual complementary network framework for graph representation learning. Two graphs representing topology and features respectively are first constructed. Then, two branches leveraging the two graphs are proposed to explore different aspects of the original graph. Finally, a diversity loss is presented to capture the rich information of node features.
To me, the overall presentation is barely satisfactory, with many proposals not well-motivated. Also, the novelty is limited given the large body of existing work on exploring dual aspects of graphs. Moreover, the experiments are not convincing.

Detailed comments:
* The reason why two branches of GConv nets are used is not clear. I am especially not clear how the proposed DGCN differs from dual-channeled GAT, why the diversity loss is not employed on attention heads, and how does the embedding learnt by GCN supplement the information of that by GAT. More elaborations are needed.
* Experiments are not convincing; the result analysis of this paper is rather superficial.
  * Since DGCN uses two branches of GConv nets, large-scale datasets are necessary to evaluate the performance and efficiency.
  * Inconsistency between GCN and kNN-GCN. It seems that on UAI2010, BlogCatelog, and Flickr kNN-GCN is significantly better than GCN, but the opposite holds for the other two datasets. It should be noted why the two methods show such different performance on different datasets.
* Given the large amount of existing literature regarding dual networks, many related methods are missing. The authors should especially pay attention to network embedding techniques, e.g., [1].

Minor:
* Mathematical expressions are in chaotic forms, which makes the readability poor.
* Page 5: CDAN -> DGCN?

[1]	Z. Meng, S. Liang, H. Bao, and X. Zhang, Co-Embedding Attributed Networks, in WSDM, 2019, pp. 393–401.

---

> ### Author Response · Authors · 2020-11-16
> **To Reviewer4**
>
> Thank you for taking the time to review our paper and we appreciate the positive and constructive feedback.
>
>
> Quote 1: “ The reason why two branches of GConv nets are used is not clear. I am especially not clear how the proposed DGCN differs from dual-channeled GAT, why the diversity loss is not employed on attention heads, and how does the embedding learnt by GCN supplement the information of that by GAT. More elaborations are needed.”
>
> The core of our model is that the networks selected by the two branches are different, so that more complementary information can be extracted. In addition, our network uses graph in the dataset and graph constructed from features to extract richer information at the same time. We use the diversity loss to expand the differences between branches, and then the attention mechanism combines the embedded information of GCN with the embedded information of GAT to obtain more abundant information.
>
>
> Quote 2: “Inconsistency between GCN and KNN-GCN. It seems that on UAI2010, BlogCatelog, and Flickr kNN-GCN is significantly better than GCN, but the opposite holds for the other two datasets. It should be noted why the two methods show such different performance on different datasets.”
>
> The difference between the two networks in different datasets should come from the use of different graphs.
> The main difference between GCN and KNN-GCN is that the structure graph and cosine graph are used respectively. Graph essentially reflects the correlation degree of nodes in a certain attribute. Taking citation network as an example, if nodes refer to papers, graph 1 thinks that nodes belonging to the same author should be connected, while graph 2 thinks that nodes with reference relationship should be connected. In the real world, there are cases where the same author writes different kinds of papers and a paper quotes different kinds of papers. In extreme cases, if on Graph 1, nodes with the same label are connected to each other;In Graph 2, nodes belonging to different classes are connected with each other. Obviously, graph 1 has a better learning effect than Graph 2. In fact, although the above situation is difficult to occur, different graph attributes obviously affect the performance of the model. Therefore, for a dataset, a graph that is more relevant to its classification goal will perform better.
>
> Quote 3: “The authors should especially pay attention to network embedding techniques.”
>
> Thank you very much for your comments, we will continue to improve in this regard.
>
>
> Quote 4: “Mathematical expressions are in chaotic forms, which makes the readability poor.
> CDAN -> DGCN?”
>
> Thank you very much for your comments. We will correct the typing errors and further improve the mathematical expressions.

---

### Official Review · AnonReviewer3 · 2020-10-27
**This paper introduces a method that combines GCN and GAT to extract features from two views of each graph for semi-supervised classification. The framework makes sense in terms of learning comprehensive features, but the method is an incremental and straightforward development on existing methods.**

**Rating:** 4
**Confidence:** 5

**Review:**

This paper introduces a method on semi-supervised graph classification. For each graph, the method first constructs another view based on the cosine similarity between nodes' features, and from the two views (topology and feature similarity), GCN and GAT are applied to extract representations. All node representations are further combined via two layers of attentions. A diversity loss that encourages dissimilarity between the learned representations of GCN and GAT is introduced to the cross-entropy loss for joint optimization. The whole framework makes sense in terms of learning meaningful node representations for classification. However, the method lacks novelty, it is an incremental development on the existing graph neural networks. The choice of GCN and GAT as the building blocks are not well justified. It is also possible to try other kinds of GNNs. The statement on GAT "ignores the inherent structure of the graph space" on page 4 is confusing since it learns weights based on the graph structures. The experimental results show the better performance of the proposed method, but are not well analyzed. It may be better to compare with other multi-graph methods such as
Yu Shi, Fangqiu Han, Xinwei He, Xinran He, Carl Yang, Jie Luo, and Jiawei Han. "mvn2vec: Preservation and collaboration in multi-view network embedding." arXiv preprint arXiv:1801.06597 (2018).
Also, it seems AM-GCN in the experiments also works on multi-view of graphs. The superiority of the proposed method compared to AM-GCN is not clearly described in the paper.

---

> ### Author Response · Authors · 2020-11-16
> **To Reviewer3**
>
> Thank you for taking the time to review our paper and we appreciate the positive and constructive feedback.
>
> Quote 1: “The choice of GCN and GAT as the building blocks are not well justified. It is also possible to try other kinds of GNNs.”
>
> We admit that branches can use other networks, but the core of our model is that the networks selected by the two branches are different, so that more complementary information can be extracted. In addition, we found that even with the same network, when we change the input, we can still get comparable results. We will submit supplementary experiments to prove the validity of our views later.
>
> Quote 2: “The statement on GAT "ignores the inherent structure of the graph space" on page 4 is confusing since it learns weights based on the graph structures. ”
>
> I'm sorry that our expression made you misunderstand: our original meaning is that compared with GCN, the degree of central node and neighbor node is more considered in aggregation. GAT adopts completely different operation in aggregation, and ignores the influence of degree attribute of node on aggregation.

---

### Decision · Program_Chairs · 2021-01-07
**Final Decision**

**Decision:**

Reject

**Comment:**

All four reviewers expressed very significant and consistent concerns on this submission during review. No reviewer is willing to support this submission during discussion. It is clear this submission does not make the bar of ICLR.